# In Vitro and In Vivo Activity of Citral in Combination with Amphotericin B, Anidulafungin and Fluconazole against *Candida auris* Isolates

**DOI:** 10.3390/jof9060648

**Published:** 2023-06-06

**Authors:** Iñigo de-la-Fuente, Andrea Guridi, Nerea Jauregizar, Elena Eraso, Guillermo Quindós, Elena Sevillano

**Affiliations:** 1Department of Immunology, Microbiology and Parasitology, Faculty of Medicine and Nursing, University of the Basque Country (UPV/EHU), 48940 Leioa, Spain; idelafuente009@ikasle.ehu.eus (I.d.-l.-F.); elena.eraso@ehu.eus (E.E.); guillermo.quindos@ehu.eus (G.Q.); elena.sevillano@ehu.eus (E.S.); 2Department of Pharmacology, Faculty of Medicine and Nursing, University of the Basque Country (UPV/EHU), 48940 Leioa, Spain; nerea.jauregizar@ehu.eus

**Keywords:** citral, drug combination, *Candida auris*, *Caenorhabditis elegans*, antifungal resistance, amphotericin B, anidulafungin, fluconazole, synergistic effect

## Abstract

*Candida auris* is an emerging fungal pathogen responsible for hospital outbreaks of invasive candidiasis associated with high mortality. The treatment of these mycoses is a clinical challenge due to the high resistance levels of this species to current antifungal drugs, and alternative therapeutic strategies are needed. In this study, we evaluated the in vitro and in vivo activities of combinations of citral with anidulafungin, amphotericin B or fluconazole against 19 *C. auris* isolates. The antifungal effect of citral was in most cases similar to the effect of the antifungal drugs in monotherapy. The best combination results were obtained with anidulafungin, with synergistic and additive interactions against 7 and 11 of the 19 isolates, respectively. The combination of 0.06 μg/mL anidulafungin and 64 μg/mL citral showed the best results, with a survival rate of 63.2% in *Caenorhabditis elegans* infected with *C. auris* UPV 17-279. The combination of fluconazole with citral reduced the MIC of fluconazole from > 64 to 1–4 μg/mL against 12 isolates, and a combination of 2 μg/mL fluconazole and 64 μg/mL citral was also effective in reducing mortality in *C. elegans*. Amphotericin B combined with citral, although effective in vitro, did not improve the activity of each compound in vivo.

## 1. Introduction

The different species of the genus *Candida* are responsible for candidiasis, the most common opportunistic mycosis suffered in humans. These microorganisms are commensal in healthy individuals as part of the gastrointestinal and vaginal microbiota and can cause various clinical manifestations of varying severity that range from mucocutaneous overgrowth to bloodstream infections, called candidemia or invasive candidiasis [1].

The main species involved in candidiasis are *Candida albicans*, *Candida glabrata*, *Candida krusei*, *Candida parapsilosis* and *Candida tropicalis.* In recent years, the global emergence of *Candida auris* has been of particular concern due to its ability to cause hospital outbreaks of invasive candidiasis. This species causes nosocomial infections with a higher mortality rate, approximately 30–60%, compared to other *Candida* species, although death is not always attributable to the infection, as patients often suffer from other serious underlying diseases [2,3]. This species has been catalogued as an emerging fungal pathogen that represents a serious global health threat [4,5]. In addition, it can persist in the hospital environment for prolonged periods of time, being a constant source of infection affecting a large number of patients usually admitted to intensive care units or post-surgical resuscitation units [3,6,7,8,9,10].

The treatment of invasive candidiasis caused by *C. auris* is a major clinical challenge due to the high level of resistance exhibited by this species. In fact, a study published by Chowdary et al. [11] revealed that 90% of 350 Indian isolates were resistant to fluconazole, 8% to amphotericin B and 2% to echinocandins. In addition, 25% showed a multidrug resistance profile. This problem highlights the necessity for searching for innovative therapies to treat these infections. One of the most promising therapies is based on the combination of antifungal drugs, such as fluconazole, with other commonly used drugs, even if they do not have antifungal activity, in order to obtain synergistic effects. This strategy, in addition to effectively killing the pathogen, avoids excessive and inappropriate use of antifungal drugs by reducing the doses used and minimizing the possible adverse effects of treatment and the ability to develop resistance [3,12,13,14,15].

The combination of antifungal drugs with other compounds has been evaluated in in vitro experiments with promising results. Among other compounds, the use of the essential oils of some plants has also been described due to their antifungal, antibacterial and/or antiparasitic activities. Some of these oils, as well as their constituents, have shown anti- *Candida* activity [16,17], as in the case of citral (3,7-dimethyl-2-6-octadienal). This compound is a terpene mixture of two geometric isomers: the geranial or trans-citral and the neral or cis-citral. They are aldehydes present in many citrus fruits and other herbal and spice essential oils, such as lemons, oranges or lemongrass [18]. The authors have also described that citral has the ability to damage the integrity of the plasma membrane through a mechanism that destabilizes the membrane itself, affecting its structure, blocking its synthesis, compromising its integrity and permeability and ultimately causing cell death [18]. Previous results obtained from in vitro and in vivo studies by our research group showed activity against other *Candida* species, such as *C. albicans*, *C. glabrata* and *C. krusei* [19].

The aim of the present work was to test antifungal drug combinations with citral, following the method of microdilution in broth, against *C. auris*, and to evaluate the most relevant combinations in vivo in the alternative animal model *Caenorhabditis elegans*, which allows the use of a high number of individuals per test. To our knowledge, this is the first study to evaluate the antifungal activity of citral in combination with current antifungal agents against *C. auris* using in vitro and in vivo models.

## 2. Materials and Methods

### 2.1. Fungal Strains

The effect of the combinations was tested against 19 clinical isolates of *C. auris* isolated from blood (6), oropharyngeal (7) and urine (6) samples at the Microbiology Service of the Hospital Universitario y Politécnico La Fe (Valencia, Spain). Reference strains obtained from the American Type Culture Collection (ATCC), *Candida parapsilosis* ATCC 22,019 and *Candida krusei* ATCC 6258, were used as quality controls in the experiments.

### 2.2. Drugs Tested

The antifungal drugs used were amphotericin B (Sigma Aldrich, Texas, USA), anidulafungin (Pfizer SA, Madrid, Spain) and fluconazole (Pfizer SA, Madrid, Spain) in monotherapy and in combination with citral (CIT) (Sigma Aldrich, Texas, USA). The final concentration of anidulafungin (ANI) and amphotericin B (AMB) ranged from 0.008 to 4 μg/mL; that of fluconazole (FLZ) from 1 to 64 μg/mL and that of citral from 2 to 1024 μg/mL, and they were prepared according to the manufacturer’s recommendations.

### 2.3. In Vitro Antifungal Susceptibility Studies

#### 2.3.1. Checkerboard Assay

The in vitro susceptibility testing of the different isolates of the combination of antifungal drugs with citral was performed in 96-well flat-bottom microtiter plates by determining the MIC according to the EUCAST guidelines modified for drug combinations in the so-called checkerboard assay (8 × 12 design) [20,21].

RPMI 1640 containing L-arginine and 2% glucose and buffered with 0.165 M morpholino propane sulfonic acid (MOPS) was used as a culture medium. After adjusting the pH to 7.0 ± 0.1, the medium was sterilized by filtration and stored at 4 °C. A stock solution of the compounds in dimethyl sulfoxide (DMSO) was prepared for the assays.

Anidulafungin or amphotericin B were added to columns 2–11 at concentrations ranging from 0.008 to 4 μg/mL. Citral was added to rows A-G, with a range of 16 to 1024 μg/mL. For the combination of fluconazole with citral, fluconazole was added to rows A-G with concentrations ranging from 1 to 64 μg/mL and citral was added to columns 2–11 with concentrations ranging from 2 to 1024 μg/mL. Fifty microliters of each drug concentration was added to the corresponding wells. Wells from column 12 were used for growth control by adding 100 μL of RPMI supplemented with 2% glucose and 2% DMSO; wells H1 and H12 were used as sterility controls.

*C. auris* isolates, previously incubated at 37 °C overnight, were suspended in API^®^ ampoules containing 0.85% NaCl medium (bioMérieux S.A., Craponne, France) to obtain a starting inoculum of 0.5–2.5 × 10^5^ CFU/mL. One hundred microliters of this suspension was inoculated in each well of the microtiter plates, excluding H1 and H12. The plates were then incubated at 37 °C for 48 h, after which the absorbance of each well was measured on an Infinite F50 spectrophotometer (Tecan, Männendorf, Switzerland) at a wavelength of 450 nm. Each experiment was performed in triplicate, with replicates carried out in independent assays.

#### 2.3.2. Data Analysis and Interpretation of Results

To evaluate the type of interactions of each combination against the tested *Candida* isolates, the results of the in vitro studies were analysed using the fractional inhibitory concentration index (FICI) based on Loewe’s theory of additivity and the response surface model based on Bliss’s theory of independence [22].

The FICI, a non-parametric approach based on Loewe’s theory, is defined as the MIC of each drug when it acts in combination divided by the MIC of the drug when it acts alone. In this way, drug interactions are classified as synergistic (FICI < 0.5), additive (FICI ≥ 0.5 but <1), indifferent (FICI ≥ 1 but <4) or antagonistic (FICI ≥ 4) [23].

In Bliss’s non-interaction theory of independence, the sum of all statistically significant synergistic and antagonistic interactions (ΣSYN_ANT) was the parameter that summarized the whole interaction surface for the studied combinations [24]. Additionally, when the ΣSYN_ANT value obtained for each checkerboard assay was lower than 100%, the interaction was defined as weak, values between 100% and 200% were defined as moderate and those higher than 200% were considered strong [25]. Combenefit was the software used to perform the Bliss analysis by means of parametric determination [26]. Combenefit creates a reference surface based on Bliss independence that is evaluated from the dose–response curves of each of the two combined agents.

### 2.4. In Vivo Assays

#### 2.4.1. Growth Conditions

The double mutant strain AU37 (*glp-4*(*bn2*); *sek-1*(*km4)*) of *C*. *elegans* and the non-pathogenic strain of *Escherichia coli* OP50 were obtained from the *Caenorhabditis* Genetics Center (University of Minnesota, USA). The AU37 double mutant strain increases the susceptibility to various pathogens (*sek-1*) and maintains a constant number of sterile worms at 25 °C (*glp-4*). Nematodes were maintained in a nematode growth medium (NGM, 2.4 g NaCl, 13.6 g bacteriological agar, 2 g peptone, 800 μL CaCl_2_ (1 M), 800 μL of 5 mg/mL cholesterol in ethanol, 800 μL MgSO_4_ (1 M), 20 mL KPO4 and 777 mL distilled H_2_O) in agar plates at 15 °C, in which there was previous non-pathogenic *E. coli* OP50 strain growth as feed [27]. The test yeast cells were cultured during the 18–24 h before the assay in yeast extract peptone dextrose liquid medium (YEPD, 1% yeast extract, 2% bacteriological peptone, 2% D-glucose) at 30 °C with shaking.

#### 2.4.2. Survival Assay in *Caenorhabditis elegans*

The assays were performed as previously described by Breger et al. [28]. Nematodes fed for 2 h at 25 °C with the different isolates of *C. auris* were subsequently transferred onto brain heart infusion (BHI) agar plates (Panreac, Barcelona, Spain). Afterward, nematodes were washed with M9 buffer (3 g KH _2_PO_4_, 6 g Na_2_HPO_4_, 5 g NaCl, 1 mL MgSO_4_ 1 M and H_2_O to 1 L) and supplemented with kanamycin (90 µg/mL) to prevent the growth of the *E. coli* OP50 strain and left on plates with NGM agar to eliminate yeast cells attached to the cuticle of the nematodes. Twenty individuals were dispensed in each well of a microtiter plate that contained M9 buffer supplemented with kanamycin and 10 μg/mL of cholesterol in ethanol. Microtiter plates were incubated at 25 °C in the dark and a visual scoring of live and dead worms was performed every 24 h for 120 h using a stereomicroscope (Nikon SMZ-745, Tokyo, Japan).

#### 2.4.3. Treatment with Antifungal Drugs in Combination with Citral

For the evaluation of the treatment efficacy, nematodes were treated with the following concentrations used in monotherapy or in combination: 0.06 and 0.25 µg/mL of anidulafungin, 0.03 and 0.25 µg/mL of amphotericin B and 1 and 2 µg/mL of fluconazole with 32, 64 and 128 µg/mL of citral, all diluted in dimethyl sulphoxide (DMSO), which was added to the assay liquid media. The plates were incubated at 25 °C in the dark and survival was verified by a visual scoring of live and dead worms every 24 h for 120 h.

#### 2.4.4. Statistics

All the in vivo experiments were performed in at least three independent assays and the nematodes used were synchronized to the L4 larval stage before starting the assay.

The results obtained in the survival and antifungal treated nematode assays were analysed by Kaplan–Meier survival curves and the differences between them were estimated by a Log-rank test with the statistical program SPSS v28.0 (IBM, Armonk, NY, USA) (*p* < 0.05 was considered statistically significant).

## 3. Results

### 3.1. Efficacy of In Vitro Combinations

In vitro experiments with 19 clinical isolates of *C. auris* with combinations of amphotericin B, anidulafungin and fluconazole with citral revealed synergistic or additive interactions for most of the clinical isolates of *C. auris* studied and for all drug combinations tested.

The MIC of amphotericin B in monotherapy against *C. auris* isolates ranged from 0.125 to 0.5 μg/mL in all cases, and the MIC of citral ranged from 64 to 1024 μg/mL (Table 1). The combination of both compounds was able to reduce the MIC values of amphotericin B two- to four-fold against 17 out of 19 isolates, while the MIC of citral was reduced from 128–1024 μg/mL to 32–64 μg/mL against 17 out of 19 isolates. Accordingly, the effect of the combination of amphotericin B with citral improved the effect of monotherapy treatment, as the FICI results showed a synergistic result against 42.11% of the isolates and an additive result against 47.37%.

The values of the summary parameter of the Bliss independence-based model, ΣSYN_ANT, showed synergistic interactions for all combinations and isolates, although weak (values less than 100%). This synergy was observed against all the isolates with combinations including high concentrations of citral ranging from 256 to 1024 μg/mL and medium concentrations of amphotericin B ranging from 0.008 to 0.0625 μg/mL, as can be seen in Figure 1 for isolate *C. auris* UPV 17-280.

The MIC of anidulafungin ranged between 0.06 and 4 μg/mL, while the MIC of citral in monotherapy was between128 and 1024 μg/mL (Table 2). Analysing the results obtained with the drugs in combination, a marked reduction in the MIC of the two compounds studied was observed compared to the MIC in monotherapy. The MIC of anidulafungin was reduced to concentrations of 0.015–0.03 μg/mL against 15 of the isolates and the MIC of citral was reduced two- to four-fold against 17 out of 19 isolates. When interpreting the results with the FICI model, we obtained a synergistic effect against 36.84% of the isolates and an additive effect against 57.89%.

The combination of anidulafungin with citral proved to be synergistic against most of the strains tested using the Bliss interaction model. This synergistic effect was observed using medium and high concentrations of anidulafungin and a wide range of citral concentrations, without observing a clear dose-dependent pattern, as can be seen in Figure 2 for a representative isolate of *C. auris*. Although the synergy obtained is mild (metric values between 34.47 and 157.8), it was observed against all tested isolates.

Regarding the combination of fluconazole and citral, all isolates showed resistance to fluconazole in monotherapy with MIC values above 64 μg/mL. On the other hand, citral’s MICs in monotherapy were in the range between 64 and 1024 μg/mL. It is remarkable that the combination of fluconazole with citral reduced the MIC of fluconazole from >64 to 1–4 μg/mL against 12 of the isolates (Table 3). The FICI model interpreted these results as synergistic against 10.53% of the isolates and as additive against 15.79% of them.

The combination of fluconazole and citral provided concentration-dependent synergism or antagonism results when observed in the Bliss interaction model. Certain concentrations of citral must be exceeded to obtain a synergistic effect, since at low concentrations, an antagonistic effect was observed. This concentration-dependent behaviour was also observed for fluconazole, since when using the highest concentration tested, 64 μg/mL, the antagonistic effect was lost, even if low concentrations of citral were used (Figure 3).

### 3.2. Efficacy of In Vivo Combinations

The efficacy of the different antifungal drugs used both in monotherapy and in combination against the in vivo *C. elegans* model infected with three isolates of *C. auris* (UPV 17-267, UPV 17-279 and UPV 17-281) is shown in Table 4, where the survival rates of the nematodes after receiving the different treatment alternatives are represented. The strains were selected on the basis of the origin, including samples from different clinical origins (oropharyngeal, blood and urine samples). In addition, they were chosen based on the results of the combinations in the in vitro experiments, including those whose interaction reflected the most representative result of each combination. For this experiment, the concentrations selected were 0.06 and 0.25 μg/mL for anidulafungin, 0.03 and 0.25 μg/mL for amphotericin B, 1 and 2 μg/mL for fluconazole, and finally, 32, 64 and 128 μg/mL for citral.

The most remarkable results of these in vivo experiments were the improvement in the survival rate of the nematode *C. elegans* when using the combination of drugs instead of monotherapy in almost all cases. In addition, it should be noted that these survival rates are similar to those obtained with citral in monotherapy.

Regarding the combinations, the nematodes infected with *C. auris* isolates UPV 17-267 and UPV 17-281 showed higher survival rates after the treatment with fluconazole or with a combination, and in the case of the isolate UPV 17-281, the difference in the effect of the combination versus monotherapy with fluconazole was already observed at 48 h, and is more remarkable at 72 and 96 h. The treatment of nematodes infected with the three strains with the combination of anidulafungin and citral resulted in an increase in the survival rate compared to treatment with anidulafungin in monotherapy, with this difference being most notable at 96 h, even when lower concentrations of anidulafungin were used in the combination. On the other hand, in the case of amphotericin B, the best survival results were obtained using each of them in monotherapy.

It should be noted that the difference in survival percentages between untreated and treated nematodes can be observed from 48 h and reaches the greatest differences after 72 h and 96 h. This can be observed when comparing the survival percentages obtained in the nematodes infected with the three strains and their subsequent treatment with fluconazole (both in monotherapy and in combination). For example, at 72 h after infection with the *C. auris* UPV 17-267 isolate, the survival rate of nematodes treated with the combination of fluconazole and citral was 59.1% compared to 20% in untreated nematodes, and at 96 h, a survival rate of 49.8% was observed in treated nematodes, compared to 7.8% in untreated nematodes.

Figure 4 shows the viability of *C. elegans* after infection with the three isolates and subsequent treatment with fluconazole, citral and their combination. The percentages of survival of the infected untreated nematodes varied slightly depending on the isolate they were infected with. When infected with the *C. auris* UPV 17-281 isolate, survival after 96 h was 35.4%, showing the highest survival percentage of the three isolates for untreated nematodes; when infected with the *C. auris* UPV 17-267 isolate, survival was 7.8%; and finally, the survival percentage of untreated nematodes infected with the *C. auris* UPV 17-279 isolate was 11.3%. The survival of nematodes infected with the three isolates and treated with both monotherapy and combination therapy was significantly higher than that of untreated nematodes (*p* < 0.0001).

Citral in monotherapy showed a greater effect than fluconazole, which could be observed from 48 h, being more remarkable at 96 h. In fact, at 96 h, 32 μg/mL citral in monotherapy improved the survival rate of the worms infected with *C. auris* UPV 17-279 up to 57.5% with respect to the untreated worms (*p* < 0.0001).

Treatment of nematodes infected with isolates *C. auris* UPV 17-267 and UPV 17-281 with a combination of 2 μg/mL of fluconazole and 64 μg/mL of citral improved the survival rate by 11.2% (*p* = 0.044) and 18.9% (*p* < 0.0001), respectively, compared to infected nematodes treated with fluconazole monotherapy at 96 h, and by 4.2% (*p* = 0.046) and 16.8% (*p* = 0.027), respectively, compared to the treatment with citral in monotherapy.

When comparing the survival rates of the worms treated with a combination of fluconazole and citral with respect to the untreated nematodes infected with the three isolates, an improvement of 33.8% to 39.8% at 72 h and of 39.1% to 51% at 96 h was observed.

Regarding the results of the survival rates when using anidulafungin and citral as treatments, we observed, as shown in Figure 5**,** that the survival rate of the nematodes using anidulafungin in monotherapy at 96 h was low, with rates between 13 and 18% when infected with *C. auris* UPV 17-267 and UPV 17-279 isolates. In contrast, the three combinations of anidulafungin with citral showed a significantly higher survival rate, with the combination of 0.06 μg/mL of anidulafungin and 64 μg/mL of citral in nematodes infected with *C. auris* UPV 17-279 showing the best results with a 63.2% survival rate. In the case of nematode infection with *C. auris* UPV 267 and *C. auris* UPV 279 isolates, treatment with all three combinations significantly increased survival compared to using the drug in monotherapy (*p* < 0.0001). In the treatment of *C. auris* UPV 17-281 infection, the combinations of 0.06 μg/mL anidulafungin and 64 μg/mL citral and 0.25 μg/mL anidulafungin and 32 μg/mL citral significantly increased nematode survival compared to the use of the drug in monotherapy (*p* = 0.001 and *p* = 0.005, respectively). However, the combination of 0.25 μg/mL anidulafungin and 128 μg/mL citral did not improve survival with respect to monotherapy (*p* = 0.576).

In any case, the three combinations analysed showed a higher effect on the infection of the three isolates than the effect observed with anidulafungin in monotherapy. Moreover, in all cases, the effect of citral as monotherapy offered survival rates close to the values obtained with the combination therapy. These observed differences were already observed at 72 h after the first treatment.

Finally, Figure 6 shows the survival results when treating the infected nematodes with amphotericin B and citral, both in monotherapy and in combination. After infection of the nematodes with the three selected isolates, the great effect of amphotericin B in monotherapy is remarkable. The combinations in this case did not offer better results (*p* > 0.05), but it should be noted that treatment with 32 μg/mL of citral in monotherapy was more effective than the treatment with 0.25 μg/mL of amphotericin when infected with the isolate *C. auris* UPV 17-279, showing a survival rate of 68.8%.

## 4. Discussion

The treatment of invasive candidiasis caused by *C. auris* or other species of *Candida* resistant to antifungal drugs is a clinical challenge that requires the search for new therapeutic strategies. One of the most promising alternatives is the combination of antifungal drugs with other compounds and drugs of different nature in order to achieve synergistic effects to overcome this antifungal resistance. In addition, the search for a compound that could be combined with fluconazole, a low-cost and non-toxic drug widely used in the treatment of candidiasis, to improve its efficacy or even allow its use against resistant isolates would be of great importance. In this way, although anidulafungin and amphotericin B have lower rates of resistance than fluconazole or other azoles, their combination with another product in order to achieve synergies would allow for a reduction in the dose and side effects [7].

In this work, we analysed the effect of terpene, a biologically active compound present in some plants composed of two isoprene units with the molecular formula C_10_H_16_. Citral is the name given to a mixture of two geometric isomers called geranial and neral. The antimicrobial properties of citral have been demonstrated against several fungi such as *Alternaria alternata*, *Aspergillus ochraceus* or *Penicillium expansum* or against bacteria such as *Staphylococcus aureus* or *Escherichia coli* [29,30,31,32,33]. One of the main findings of this work is the antifungal effect shown by citral in both in vitro and in vivo experiments, which in many cases is similar to the activity of antifungal drugs in monotherapy.

In the present work, the combination of fluconazole with citral proved to be synergistic against two of the nineteen *C. auris* isolates and additive against three of them. In addition, it should be noted that the main advantage lies in the reduction in the MIC of the antifungal agent required when used in combination with citral, with a reduction in the MIC of fluconazole from >64 µg/mL to 1 µg/mL. This raises the possibility of continuing to use fluconazole, a cheap, safe and easy to administer antifungal drug, against resistant isolates.

When comparing the results with those obtained in other studies, it is worth mentioning the work of Miranda-Cadena et al. [19], in which they observed a synergistic effect against seven isolates (63.63%) of *Candida* species when using combinations of citral and fluconazole. These differences between the two works can be attributed to the fact that in the case of Miranda-Cadena et al., they analysed the effect against *C. albicans*, *C. dubliniensis*, *C. glabrata*, *C. krusei*, *C. parapsilosis* and *C. tropicalis*. This reinforces and demonstrates the intrinsic resistance of the *C. auris* species to the action of antifungal agents.

When analysing the combinations with the Bliss method, we observed that, depending on the concentrations of citral used, the results were synergistic (medium to high concentrations) or antagonistic (low concentrations). This concentration-dependent effect has also been described by other authors, such as Meletiadis et al. [34], when using azoles in combination with amphotericin B, which reinforces the need to select concentrations accurately.

The best combination results with citral, including synergistic and additive effects, were obtained with anidulafungin, which showed synergistic activity against seven of the isolates and additive activity against eleven. Due to the increasing reports of resistant *C. auris* isolates, the availability of such an alternative represents an interesting treatment option against candidiasis caused by *C. auris* since, in addition to the enhancement of the synergistic effect, a reduction in the MIC of anidulafungin was observed. This means that it can be used at lower doses, thus avoiding the emergence of resistant isolates. We observed that the effect of the combination of anidulafungin and citral is dose dependent, with intermediate concentrations of anidulafungin being more effective. This is consistent with the results observed by other authors, who detected that the effects if anidulafungin in combination with drugs such as fluconazole may also be dose dependent, with a stronger effect observed at intermediate concentrations [35].

The combination of citral with amphotericin B also revealed interesting results, showing a synergistic effect against eight of the *C. auris* isolates and an additive effect against nine. In the work of O’Brien et al. [36], which analysed the effect of the combination of amphotericin B with epoxy-amine oligomers from terpenes against one isolate of *C. albicans* and one isolate of *Trichoderma virens*, they also observed a synergistic effect.

It is worth noting again the difference in the type of species tested, which again highlights the difficulty of treating infections caused by *C. auris* species. Thus, for example, in the work of Dudiuk et al. [37], a four times higher concentration of amphotericin B was required to reduce the growth of 99.9% of the initial inoculum of *C. auris* than for *C. guillermondii*. Therefore, it should be emphasized that the combination of amphotericin B with citral has a synergistic effect against *C. auris*, achieving a two- to four-fold reduction in amphotericin B MICs when combined with citral.

It is of great importance to test whether the promising results observed in vitro are maintained in in vivo experiments. In our case, we used the *C. elegans* animal model which has proven to be a useful infection model for studying host–microbe interactions of many fungal and bacterial pathogens, including the genus *Candida*. The *C. elegans* infection protocol is relatively simple and is often carried out by substituting its standard laboratory food, *E. coli*, for the desired pathogen, which colonises the *C. elegans* gut. In addition, the innate immune system of *C. elegans* includes the SEK-1 protein, a MAP kinase (MAPKKK) that is homologous to the MKK3/6 and MKK4 family of mammalian MAPKs, which activates the *C. elegans* p38 MAP kinase orthologue. This pathway has been proposed to be an ancient and conserved component of *C. elegans’* immune response to pathogens. As such, the mutation in *sek*-1 increases its susceptibility to microbial colonisation, including infection by many *Candida* species, and is required to induce a proper antifungal immune defence [38,39].

The mortality of *C. elegans* when infected with the three selected *C. auris* isolates was very high, with survival rates ranging from 7.8% to 35.4%. Our results were similar to those observed by Hernando-Ortiz et al. [38], who obtained a survival rate of less than 20% after 120 h when infected with *C. auris* isolates. In another work carried out also by Hernando Ortiz et al. [39], echinocandins were again the drugs that achieved the best survival results, although in this case the species used to infect *C. elegans* were *C. glabrata* and *C. nivariensis*. This is in agreement with our results, in which the combination that offered the best results was the one of 0.06 μg/mL anidulafungin and 64 μg/mL citral, since when treating the infected nematodes with the isolate *C. auris* UPV 17-281, the survival increased to 79.6%. It is also noteworthy that in the work carried out by Hernando Ortiz et al., an anidulafungin concentration of 8 μg/mL was used, while in the present work, when combining this drug with citral, the effect was achieved with only 0.06 μg/mL of anidulafungin.

As observed in the in vitro assay, the combination of 2 μg/mL of fluconazole and 64 μg/mL of citral was also found to be effective in reducing mortality in the nematode *C. elegans*. Although the exact mechanism by which citral enhances the effect of fluconazole is not known, it makes sense, as proposed by Miranda-Cadena et al. [19], that citral allows the antifungal agent to penetrate the cells and thus achieve greater activity. These results with fluconazole differ from those obtained in the study of Hernando-Ortíz et al. [40], since they required the highest doses of fluconazole (64 or 128 μg/mL) to detect a significant increase in the survival of *C. elegans* infected with *C. glabrata* or *C. nivariensis*.

Amphotericin B in monotherapy showed good results, with a survival rate of 63%. This was also observed in the work of Dudiuk et al. [37], in which they analysed the effect of both amphotericin B and anidulafungin against different isolates of *C. auris* and showed the efficacy of both drugs. In any case, it should be noted that these survival rates are similar to those obtained with citral in monotherapy, which reinforces the antifungal potential of this compound, although the combination did not improve the activity of each individual compound.

Although it is acknowledged that it is not possible to control the infectious dose in *C. elegans*, this model has allowed us to obtain favorable results that serve as a first step for further studies. In this sense, it would be interesting to complement these studies with other in vivo models in which the infective dose can be controlled, as in the case of *Galleria mellonella* or the murine model.

In conclusion, the combinations of amphotericin B, anidulafungin and fluconazole with citral showed promising results in susceptibility studies against clinical isolates of *C. auris*, as synergistic and additive effects were detected with all the combinations. The in vivo experiments confirmed these results, as higher survival rates of *C. elegans* were observed when using combinations of citral with anidulafungin and fluconazole. In all the combinations, the use of citral contributed to a reduction in the MIC of the antifungal drugs tested. Although further cytotoxicity assays would be necessary to ensure the safety of this compound, these results are highly promising, as the treatment of *C. auris* infections is often a clinical challenge, for which new therapeutic options are needed.

## Figures and Tables

**Figure 1 jof-09-00648-f001:**
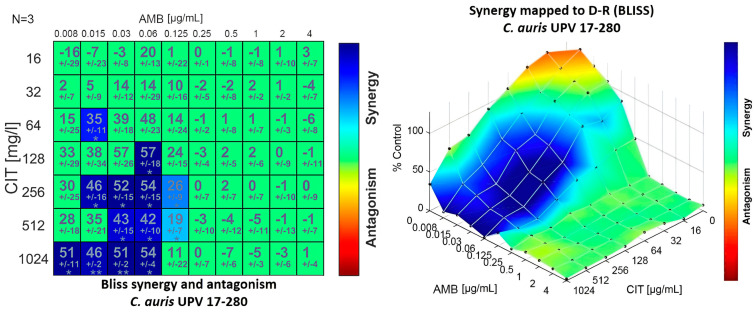
Synergy distribution determined by a Bliss interaction model for the combination of amphotericin B (AMB) and citral (CIT) against *C. auris* 17-280. Left: matrix synergy plot with synergy scores for each combination. Right: synergy distribution mapped to the dose–response surface. (* *p* < 0.05; ** *p* < 0.001).

**Figure 2 jof-09-00648-f002:**
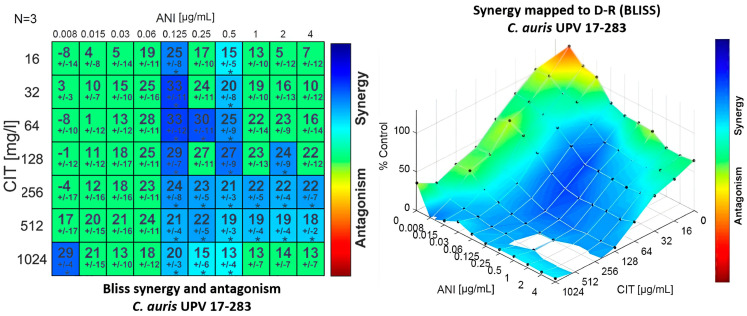
Synergy distribution determined by a Bliss interaction model for the combination of anidulafungin (ANI) and citral (CIT) against *C. auris* UPV 17-283. Left: matrix synergy plot with synergy scores for each combination. Right: synergy distribution mapped to the dose–response surface. (* *p* < 0.05).

**Figure 3 jof-09-00648-f003:**
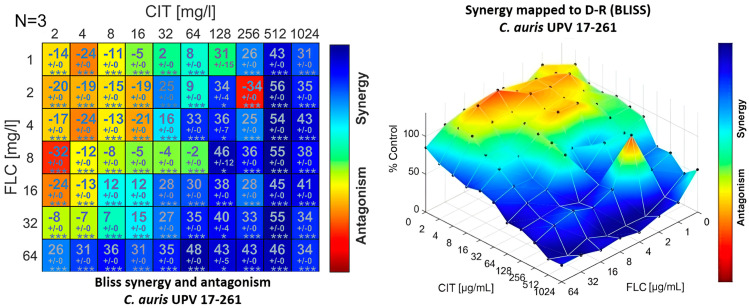
Synergy distribution determined by a Bliss interaction model for the combination of fluconazole (FLZ) and citral (CIT) against *C. auris* UPV 17-261. Left: matrix synergy plot with synergy scores for each combination. Right: synergy distribution mapped to the dose–response surface. (* *p* < 0.05; *** *p* < 10^−4^).

**Figure 4 jof-09-00648-f004:**
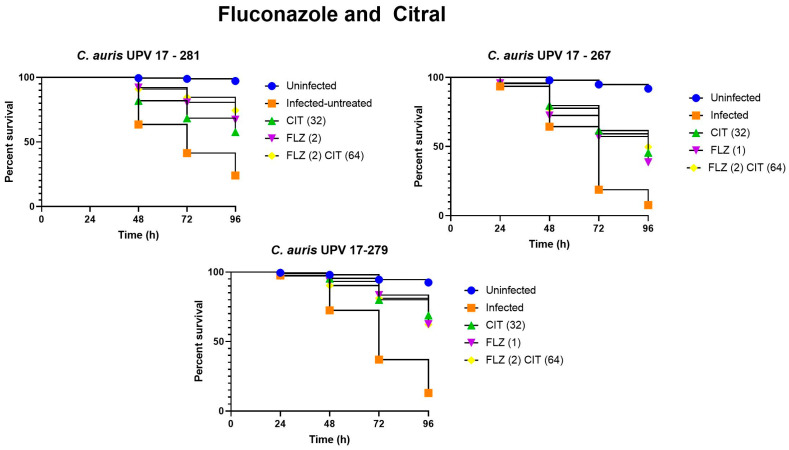
Survival assay of *C. elegans* infected with isolates *C. auris* UPV 17-281, *C. auris* UPV 17-267 and *C. auris* UPV 17-279 and treated with fluconazole (FLZ) and citral (CIT).

**Figure 5 jof-09-00648-f005:**
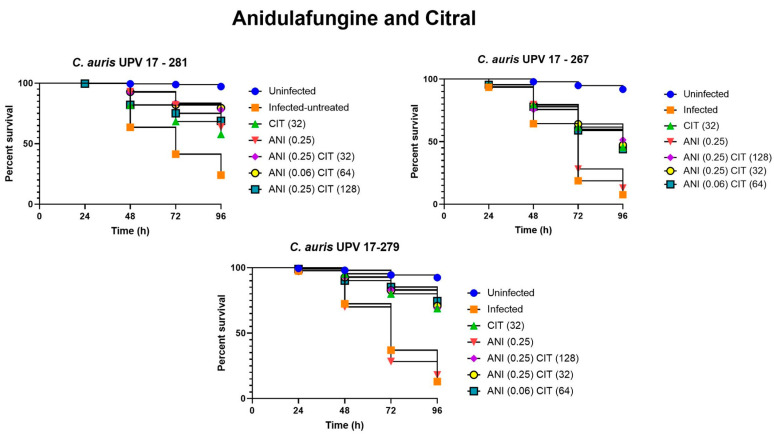
Survival assay of *C. elegans* infected with *C. auris* UPV 17-281, *C. auris* UPV 17-267 and *C. auris* UPV 17-279 isolates and treated with anidulafungin (ANI) and citral (CIT).

**Figure 6 jof-09-00648-f006:**
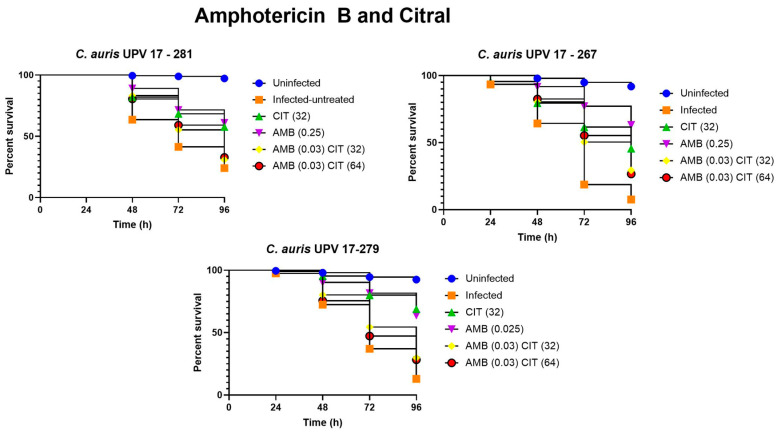
Survival assay of *C. elegans* infected with *C. auris* UPV 17-281, *C. auris* UPV 17-267 and *C. auris* UPV 17-279 isolates and treated with amphotericin B (AMB) and citral (CIT).

**Table 1 jof-09-00648-t001:** In vitro antifungal activity of the combination of amphotericin B (AMB) with citral (CIT) against *C. auris* isolates.

	MIC (μg/mL)					
	Monotherapy	Combination				
Isolates	AMB	AMB	CIT	FICI	Interpretation	ΣSYN ANT
*C. auris* UPV 17-213	0.125	0.06	32	0.54	AD	21.00
*C. auris* UPV 17-259	0.125	0.06	64	0.73	AD	53.20
*C. auris* UPV 17-261	0.25	0.015	128	1.06	IND	24.96
*C. auris* UPV 17-265	0.25	0.03	64	0.62	AD	45.35
*C. auris* UPV 17-267	0.125	0.03	64	0.30	SYN	44.19
*C. auris* UPV 17-269	0.125	0.03	64	0.74	AD	19.77
*C. auris* UPV 17-270	0.25	0.03	64	0.62	AD	64.36
*C. auris* UPV 17-272	0.25	0.015	64	0.56	AD	46.08
*C. auris* UPV 17-274	0.125	0.03	64	0.49	SYN	70.48
*C. auris* UPV 17-276	0.125	0.015	64	0.62	AD	69.86
*C. auris* UPV 17-278	0.125	0.03	64	1.24	IND	39.00
*C. auris* UPV 17-279	0.125	0.03	64	0.74	AD	36.16
*C. auris* UPV 17-280	0.125	0.03	64	0.49	SYN	71.85
*C. auris* UPV 17-281	0.125	0.03	64	0.49	SYN	65.64
*C. auris* UPV 17-283	0.25	0.03	64	0.37	SYN	53.88
*C. auris* UPV 17-285	0.125	0.03	64	0.30	SYN	59.56
*C. auris* UPV 17-289	0.125	0.03	64	0.36	SYN	51.55
*C. auris* UPV 17-291	0.125	0.03	32	0.27	SYN	69.95
*C. auris* UPV 18-029	0.5	0.125	32	0.5	AD	84.62
*C. krusei* ATCC 6258	0.031	0.015	128	1.48	IND	
*C. parapsilosis* ATCC 22019	0.06	0.015	128	0.75	AD	

FICI: fractional inhibitory concentration index; ΣSYN_ANT: total sum of synergic and antagonistic interactions; AD: additive interaction; IND: indifferent interaction: SYN: synergic interaction; AMB: amphotericin B; CIT: citral.

**Table 2 jof-09-00648-t002:** In vitro antifungal activity of the combination of anidulafungin (ANI) with citral (CIT) against *C. auris* isolates.

	MIC (μg/mL)					
	Monotherapy	Combination				
Isolates	ANI	ANI	CIT	FICI	Interpretation	ΣSYN ANT
*C. auris* UPV 17-213	1	0.03	128	0.28	SYN	38.76
*C. auris* UPV 17-259	0.06	0.015	128	0.50	AD	62.68
*C. auris* UPV 17-261	0.125	0.03	128	0.49	SYN	67.98
*C. auris* UPV 17-265	0.06	0.015	256	0.38	SYN	72.46
*C. auris* UPV 17-267	0.06	0.03	128	0.77	AD	48.59
*C. auris* UPV 17-269	0.5	0.03	256	0.56	AD	82.57
*C. auris* UPV 17-270	0.06	0.015	256	0.75	AD	157.79
*C. auris* UPV 17-272	0.06	0.03	128	0.63	AD	41.10
*C. auris* UPV 17-274	0.06	0.015	128	0.50	AD	38.74
*C. auris* UPV 17-276	0.06	0.03	128	0.56	AD	36.90
*C. auris* UPV 17-278	0.125	0.03	64	0.27	SYN	68.82
*C. auris* UPV 17-279	0.06	0.03	256	0.75	AD	53.53
*C. auris* UPV 17-280	0.125	0.03	256	0.37	SYN	54.71
*C. auris* UPV 17-281	0.125	0.06	64	0.54	AD	47.32
*C. auris* UPV 17-283	>4	0.25	64	0.16	IND	92.56
*C. auris* UPV 17-285	0.06	0.015	128	0.50	AD	34.47
*C. auris* UPV 17-289	0.06	0.03	64	0.75	AD	49.22
*C. auris* UPV 17-291	0.5	0.06	64	0.25	SYN	48.67
*C. auris* UPV 18-029	2	0.125	32	0.13	SYN	61.49
*C. krusei* ATCC 6258	1	0.5	512	0.75	AD	
*C. parapsilosis* ATCC 22019	1	0.5	128	0.56	AD	

FICI: fractional inhibitory concentration index; ΣSYN_ANT: total sum of synergic and antagonistic interactions; AD: additive interaction; IND: indifferent interaction: SYN: synergic interaction; ANI: anidulafungin; CIT: citral.

**Table 3 jof-09-00648-t003:** In vitro antifungal activity of the combination of fluconazole (FLZ) with citral (CIT) against *C. auris* isolates.

	MIC (μg/mL)					
	Monotherapy	Combination				
Isolates	FLZ	FLZ	CIT	FICI	Interpretation	Σ SYN ANT
*C. auris* UPV 17-213	>64	>64	>1024	2	IND	−113.51
*C. auris* UPV 17-259	>64	1	256	1.01	IND	-9.22
*C. auris* UPV 17-261	>64	1	128	0.07	SYN	80.86
*C. auris* UPV 17-265	>64	4	128	0.53	AD	−26.35
*C. auris* UPV 17-267	>64	1	256	0.51	AD	−41.02
*C. auris* UPV 17-269	>64	>64	>1024	2	IND	−12.32
*C. auris* UPV 17-270	>64	1	128	1.01	IND	58.75
*C. auris* UPV 17-272	>64	32	128	1.25	IND	−39.54
*C. auris* UPV 17-274	>64	2	128	1.01	IND	−52.44
*C. auris* UPV 17-276	>64	>64	>1024	2	IND	−23.68
*C. auris* UPV 17-278	>64	2	128	1.01	IND	−27.80
*C. auris* UPV 17-279	>64	4	512	1.03	IND	−37.22
*C. auris* UPV 17-280	>64	32	128	0.75	AD	25.13
*C. auris* UPV 17-281	>64	>64	>1024	2	IND	−5.91
*C. auris* UPV 17-283	>64	>64	>1024	2	IND	−36.22
*C. auris* UPV 17-285	>64	2	128	1.01	IND	−17.77
*C. auris* UPV 17-289	>64	4	128	1.01	IND	−27.17
*C. auris* UPV 17-291	>64	2	64	1.01	IND	29.63
*C. auris* UPV 18-029	>64	1	256	0.13	SYN	−51.85
*C. krusei* ATCC 6258	64	1	128	1.01	IND	
*C. parapsilosis* ATCC 22019	64	1	128	0.51	AD	

FICI: fractional inhibitory concentration index; ΣSYN_ANT: total sum of synergic and antagonistic interactions; AD: additive interaction; IND: indifferent interaction: SYN: synergic interaction; FLZ: fluconazole; CIT: citral.

**Table 4 jof-09-00648-t004:** Percentage survival of nematodes treated with different concentrations of citral (CIT) in combination with fluconazole (FLZ), anidulafungin (ANI) and amphotericin B (AMB).

	Survival (%)											
	Time (h)											
	24			48			72			96		
Treatments (μg/mL)	A	B	C	A	B	C	A	B	C	A	B	C
Uninfected	100	99.5	100	98	98	99.4	94.9	94.4	98.3	92.4	92.5	97.2
Infected-untreated	92.6	95.4	100	66.9	77.1	83.8	20	41.3	49	7.8	11.3	35.4
CIT (32)	96	99	100	81.4	95.3	81.6	64.3	80.1	68.1	45.6	68.8	57.7
FLC (1)	95.8	97.3	100	72.4	93.2	84.4	57.3	83.3	63.3	38.6	62.5	56.6
FLC (2) Citral (64)	96	99	100	77.6	90.3	89.7	59.1	81.1	82.8	49.8	62.3	74.5
ANI (0.25)	100	100	100	79.5	70.1	94	27.6	28	83.8	13	18.1	63.9
ANI (0.25) CIT (128)	92.9	97.9	99.6	75.5	92.9	84.4	59.5	83.4	77.4	51.4	68.7	68.9
ANI (0.25) CIT (32)	93.4	97.5	100	80.5	92.7	92.2	64.8	82.4	82.8	47.4	70.8	78
ANI (0.06) CIT (64)	95.5	98.9	100	78	90.9	92.7	59	85.7	82.4	44	74.5	79.6
AMB (0.25)	100	100	100	91.6	90.2	81.4	76.9	81.7	66.7	63.2	63.8	61
AMB (0.03) CIT (64)	100	100	100	80.8	79.4	86.5	56.2	53.9	75	26.7	29.8	32.9
AMB (0.03) CIT (32)	100	100	100	80.3	75.6	82.2	49.9	45.7	54	29.4	28.3	31

A: *C. auris* UPV 17-267; B: *C. auris* UPV 17-279; C: *C. auris* UPV 17-281. Data represent the mean of at least three independent assays.

## Data Availability

Not applicable.

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
