# Peer review of "In Vitro and In Vivo Activity of Citral in Combination with Amphotericin B, Anidulafungin and Fluconazole against Candida auris Isolates"

_jof, 2023, doi:10.3390/jof9060648_

Round 1

Reviewer 1 Report

Overall, this study is interesting.
It brings elements on the problems of:
- resistance to antifungal drugs;
- the individualized treatment of fungemia with the interest of a correct identification of the species.

Section "materials and methods". Some methodological points need to be clarified.

In the introduction the authors focused on different diseases related to candida, it would be interesting to develop the type of sampling and several clinicals information.

Result section:

Figure 4, 5 and 6 (survival assays): The statistically significant differences between the curves do not appear in the document. Can you illustrate them on the figures or in a table (in supplementary data section)?

Table 1, 2 and 3: The in vitro antifungal activity of citral in monotherapy although mentioned in the results does not appear. Can you display this result in these tables? As it is, it does not allow to correctly appreciate the effect of the combination of the two molecules.

Lines 183-185: “The combination of both compounds succeeded in reducing the MIC values of amphotericin B by two to four-fold against 17 out of 19 isolates”.

I don’t understand this paragraph ecause in table 1, I can count 13 strains with a reduced MIC of 2 or 4-fold, 3 and 1 strains with a reduced MIC of 8 and 16-fold, respectively. The two remaining show indifferent interactions.

Line 231-237: In vivo combinations essays:

I would like you to describe the reasons for the choice of these three strains. Same remarks for the combinations of concentrations tested. The Bliss synergy and antagonism distribution are not highlighted at this stage.

Line 293-295: “In fact, at 96 h, 2 μg/ml citral in mono-therapy improved up to 57.5% the survival rate of the worms infected with C. auris UPV 17-279, with respect to the untreated worms”. I don’t find this citral concentration in this essay.

The references used are current and relevant. Line58, I do not see what the reference 14 brings in this context.

The discussion section seems to be correctly treated but I need some details on the results to be sure.

Author Response

Answers are marked with *

* Thank you very much for the comments. In this new version of the manuscript, we have included the information and modifications requested by the reviewers. Responses to the comments of reviewers are given in a point-by-point manner, below:

Reviewer 1

Overall, this study is interesting.
It brings elements on the problems of:
- resistance to antifungal drugs;
- the individualized treatment of fungemia with the interest of a correct identification of the species.

* We thank the reviewer for his thorough analysis, which allowed us to detect some mistakes and include more accurate information, resulting in an improvement of the manuscript.

Section "materials and methods". Some methodological points need to be clarified. In the introduction the authors focused on different diseases related to candida, it would be interesting to develop the type of sampling and several clinicals information.

* We appreciate the reviewer's comment, we have included more information on the clinical origin of the samples in this section, that now is written as follows:

“The effect of the combinations was tested against 19 clinical isolates of C. auris isolated from blood (6), oropharyngeal (7) and urine (6) samples at the Microbiology Service of the Hospital Universitario y Politécnico La Fe (Valencia, Spain)”.

Result section:

Figure 4, 5 and 6 (survival assays): The statistically significant differences between the curves do not appear in the document. Can you illustrate them on the figures or in a table (in supplementary data section)?

* Following the reviewer's recommendations, we have included the results of the statistical analysis in the text.

Table 1, 2 and 3: The in vitro antifungal activity of citral in monotherapy although mentioned in the results does not appear. Can you display this result in these tables? As it is, it does not allow to correctly appreciate the effect of the combination of the two molecules.

* The aim of this work was to analyses the effect of citral as a boosting agent for the different antifungal drugs, whose effect is proved, and not to administer citral in monotherapy. For this reason, we have included the results of citral in the text rather than in the tables, so that they are not tedious and reflect the most relevant information.

Lines 183-185: “The combination of both compounds succeeded in reducing the MIC values of amphotericin B by two to four-fold against 17 out of 19 isolates”. I don’t understand this paragraph because in table 1, I can count 13 strains with a reduced MIC of 2 or 4-fold, 3 and 1 strains with a reduced MIC of 8 and 16-fold, respectively. The two remaining show indifferent interactions.

* We appreciate the accurate review. The reviewer is right about the number of indifferent interactions. Regarding the reduction in the MIC, in our case, we only refer to the reduction of the MIC of amphotericin B.

Line 231-237: In vivo combinations essays:

I would like you to describe the reasons for the choice of these three strains. Same remarks for the combinations of concentrations tested. The Bliss synergy and antagonism distribution are not highlighted at this stage.

* The reviewer is right. Now we have included information of the selection of the strains on the text, which now appears as follows:

“The efficacy of the different antifungal drugs used both in monotherapy and in combination against the in vivo C. elegans model infected with three isolates of C. auris (UPV 17-267, UPV 17-279 and UPV 17-281) is shown in Table 4, where the survival rates of the nematodes after receiving the different treatment alternatives is represented. The strains were selected on the basis of the origin, including samples from different clinical origin (oropharyngeal, blood and urine samples). In addition, they were chosen on the basis of the results of the combinations in the in vitro experiments, including those whose interaction reflected the most representative result of each combination. For this experiment, the concentrations selected where: 0.06 and 0.25 μg/ml for anidulafungin, 0.03 and 0.25 μg/ml for amphotericin B, 1 and 2 μg/ml for fluconazole, and finally, 32, 64 and 128 μg/ml for citral”.

Line 293-295: “In fact, at 96 h, 2 μg/ml citral in mono-therapy improved up to 57.5% the survival rate of the worms infected with C. auris UPV 17-279, with respect to the untreated worms”. I don’t find this citral concentration in this essay.

* The reviewer is right. The concentration used is 32 μg/ml citral. We have corrected this mistake in the manuscript.

The references used are current and relevant. Line58, I do not see what the reference 14 brings in this context.

* We are very sorry. We noticed this error in this reference and we thought we had corrected it. We have removed this reference from the manuscript.

The discussion section seems to be correctly treated but I need some details on the results to be sure.

* I hope that the changes included in the manuscript will clear up any doubts.

Reviewer 2 Report

The authors submitted a manuscript regarding the design of a more effective therapeutic stratgey to fight candida fungi. The manuscript is well writen and only a few things need to be modified and clarified.

Overall the document in vitro and in vivo must be in italic

Line 346 - The sentence "In this work we analysed the effect of a terpene, a biologically active plant-based 346 substance formed by the combination of five carbons called isoprene (C5H8)." is not properly written. the terpene is composed of TWO unitd of isoprene. This must be clarified in the manuscript. 

As a reader, I have question regarding the final application of these combinations. Are these for oral administration? Are they for topical application? Either way, cytotocity assays must be considered. 

Other that these I believe the manuscript is ready for publication. 

Author Response

Answers are marked with *

* Thank you very much for the comments. In this new version of the manuscript, we have included the information and modifications requested by the reviewers. Responses to the comments of reviewers are given in a point-by-point manner, below:

Reviewer 2

The authors submitted a manuscript regarding the design of a more effective therapeutic stratgey to fight candida fungi. The manuscript is well writen and only a few things need to be modified and clarified.

* We are grateful for the critical reading of the manuscript, which enabled us to improve it and prepare this new version.

Overall, the document in vitro and in vivo must be in “italic”.

* Thank you very much for the comment. Following the recommendations of the reviewer, we have modified the words that should be written in italics in the text.

Line 346 - The sentence "In this work we analysed the effect of a terpene, a biologically active plant-based 346 substance formed by the combination of five carbons called isoprene (C5H8)." is not properly written. the terpene is composed of TWO units of isoprene. This must be clarified in the manuscript. 

* As mentioned by the referee we have rewritten the text in relation to the composition of the citral, and it is now as follows:

“In this paper we analyse the effect of the terpene, a biologically active compound present in some plants, composed of two isoprene units with the molecular formula C10H16. Citral is the name given to a mixture of two geometric isomers called geranial and neral.”

As a reader, I have question regarding the final application of these combinations. Are these for oral administration? Are they for topical application? Either way, cytotoxicity assays must be considered. 

* We thank the reviewer for the comments. We are considering the study of cytotoxicity in a forthcoming work analysing the effect on murine macrophage-like cell line RAW 264.7 of both the compound without encapsulation and using different types of nanoparticles. As a result of these studies, we will have a clearer picture of the possible ways in which the drug could be administered.

* We have included in the discussion in the final paragraph that cytotoxicity testing would be necessary to ensure its safety, that now it is written as follows:

“In conclusion, the combinations of amphotericin B, anidulafungin and fluconazole with citral showed promising results in the susceptibility studies against clinical isolates of C. auris, as synergistic and additive effects were detected with all the combinations. The in vivo experiments confirmed these findings, as higher survival rates of C. elegans were observed when using the combinations of citral with anidulafungin and fluconazole. In all the combinations, the use of citral contributed to the reduction of the MIC of the tested antifungal drugs. Although further cytotoxicity assays would be necessary to ensure the safety of this compound, these results are highly promising, as the treatment of C. auris infections is often a clinical challenge, for which new therapeutic options are needed”.

Other that these I believe the manuscript is ready for publication. 

Reviewer 3 Report

In this research article entitled " In vitro and in vivo activity of citral in combination with am- photericin B, anidulafungin and fluconazole against Candida auris isolates" the authors evaluated the in vitro and in vivo activities of the combination of citral with anidulafungin, amphotericin B or fluconazole against 19 C. auris isolates. Quantitatively, there were performed enough experiments and results and discussion were presented and analyzed well in most cases. Tables and figures are mainly clear and organized. The report is interesting, because nowadays we really need substitutes for antifungal drugs, especially due to the increasing resistance of microorganisms to currently used drugs, that's why I rate this article as highly beneficial. In addition, this new resistant strain of yeast is dangerous, especially from the point of view of the low number of antifungal drugs available.

However, I mention below some points that should be considered before processing further.

-section 2.1: all latin name of microorganisms must be write in italic

- also in vitro and in vivo please must be written in italic through the document

- L215 why authors used bold? Also, some links to images and tables are in bold, please fix that.

- I recommend the authors to write the conclusion as a separate chapter, better specify the obtained results and provide the limits of this study as well as future aspects.

I recommend minor editing of the English language, but the text is easy to read and in my opinion it is good written.

Author Response

Answers are marked with *

* Thank you very much for the comments. In this new version of the manuscript, we have included the information and modifications requested by the reviewers. Responses to the comments of reviewers are given in a point-by-point manner, below:

Reviewer 3

In this research article entitled " In vitro and in vivo activity of citral in combination with am photericin B, anidulafungin and fluconazole against Candida auris isolates" the authors evaluated the in vitro and in vivo activities of the combination of citral with anidulafungin, amphotericin B or fluconazole against 19 C. auris isolates. Quantitatively, there were performed enough experiments and results and discussion were presented and analyzed well in most cases. Tables and figures are mainly clear and organized. The report is interesting, because nowadays we really need substitutes for antifungal drugs, especially due to the increasing resistance of microorganisms to currently used drugs, that's why I rate this article as highly beneficial. In addition, this new resistant strain of yeast is dangerous, especially from the point of view of the low number of antifungal drugs available.

* We appreciate the reviewer's comments and share the concern about the difficulty of treating infections caused by this new species.

However, I mention below some points that should be considered before processing further.

-section 2.1: all latin name of microorganisms must be write in italic

- also ”in vitro” and “in vivo” please must be written in italic through the document

* Thank you for the comment. Following the recommendations, we have modified the words that should be written in italics in the text.

- L215 why authors used bold? Also, some links to images and tables are in bold, please fix that.

* Thank you for the comment. Following the recommendations, we have removed the bold type where it was not necessary.

I recommend the authors to write the conclusion as a separate chapter, better specify the obtained results and provide the limits of this study as well as future aspects.

* Following the instructions of the journal, we have not given a specific title for the conclusions section. However, we have included the conclusions in a separate paragraph and following the reviewer's indications we have included limitations and future perspectives, that now it is written as follows:

“In conclusion, the combinations of amphotericin B, anidulafungin and fluconazole with citral showed promising results in the susceptibility studies against clinical isolates of C. auris, as synergistic and additive effects were detected with all the combinations. The in vivo experiments confirmed these findings, as higher survival rates of C. elegans were observed when using the combinations of citral with anidulafungin and fluconazole. In all the combinations, the use of citral contributed to the reduction of the MIC of the tested antifungal drugs. Although further cytotoxicity assays would be necessary to ensure the safety of this compound, these results are highly promising, as the treatment of C. auris infections is often a clinical challenge, for which new therapeutic options are needed”.

Comments on the Quality of English Language

I recommend minor editing of the English language, but the text is easy to read and in my opinion it is good written.

* We have revised the text and improved the English edition in some paragraphs and therefore we consider that the quality of the manuscript has improved.

Round 2

Reviewer 1 Report

I am satisfied with the explanations provided by the authors and the revisions made to the manuscript.